# The Role of Mitochondrial Fat Oxidation in Cancer Cell Proliferation and Survival

**DOI:** 10.3390/cells9122600

**Published:** 2020-12-04

**Authors:** Matheus Pinto De Oliveira, Marc Liesa

**Affiliations:** 1Department of Medicine, Division of Endocrinology, David Geffen School of Medicine at UCLA, Los Angeles, CA 90095, USA; mpoliveira@ucla.edu; 2Department of Molecular and Medical Pharmacology, David Geffen School of Medicine at UCLA, Los Angeles, CA 90095, USA; 3Molecular Biology Institute at UCLA, Los Angeles, CA 90095, USA

**Keywords:** mitochondria, fatty acid oxidation, glycolysis, lipogenesis, cancer, ISR, ATF4

## Abstract

Tumors remodel their metabolism to support anabolic processes needed for replication, as well as to survive nutrient scarcity and oxidative stress imposed by their changing environment. In most healthy tissues, the shift from anabolism to catabolism results in decreased glycolysis and elevated fatty acid oxidation (FAO). This change in the nutrient selected for oxidation is regulated by the glucose-fatty acid cycle, also known as the Randle cycle. Briefly, this cycle consists of a decrease in glycolysis caused by increased mitochondrial FAO in muscle as a result of elevated extracellular fatty acid availability. Closing the cycle, increased glycolysis in response to elevated extracellular glucose availability causes a decrease in mitochondrial FAO. This competition between glycolysis and FAO and its relationship with anabolism and catabolism is conserved in some cancers. Accordingly, decreasing glycolysis to lactate, even by diverting pyruvate to mitochondria, can stop proliferation. Moreover, colorectal cancer cells can effectively shift to FAO to survive both glucose restriction and increases in oxidative stress at the expense of decreasing anabolism. However, a subset of B-cell lymphomas and other cancers require a concurrent increase in mitochondrial FAO and glycolysis to support anabolism and proliferation, thus escaping the competing nature of the Randle cycle. How mitochondria are remodeled in these FAO-dependent lymphomas to preferably oxidize fat, while concurrently sustaining high glycolysis and increasing de novo fatty acid synthesis is unclear. Here, we review studies focusing on the role of mitochondrial FAO and mitochondrial-driven lipid synthesis in cancer proliferation and survival, specifically in colorectal cancer and lymphomas. We conclude that a specific metabolic liability of these FAO-dependent cancers could be a unique remodeling of mitochondrial function that licenses elevated FAO concurrent to high glycolysis and fatty acid synthesis. In addition, blocking this mitochondrial remodeling could selectively stop growth of tumors that shifted to mitochondrial FAO to survive oxidative stress and nutrient scarcity.

## 1. The Randle Cycle: A Hormone Independent Mechanism Linking Nutrient Availability to Anabolism and Catabolism

The Randle cycle is determined by a set of enzymes and metabolites that establish a competition between glycolysis and mitochondrial fatty acid oxidation (FAO). Randle and colleagues demonstrated that higher availability of extracellular glucose was sufficient to increase glycolysis and suppress FAO in isolated hearts [1]. Closing this cycle, higher availability of extracellular fatty acids (FA) increased FAO and blocked glycolysis in a normal heart, without changing the ATP/ADP ratio [1]. Randle, Garland, and colleagues found that FAO decreased the activity of the cytosolic glycolytic enzymes that preceded pyruvate synthesis, through a multi-step process involving mitochondrial and cytosolic reactions (Figure 1A) [2,3]. The first step in this process is the generation of acetyl-CoA and NADH in the mitochondria, which are the final products of FAO. Remarkably, glucose-derived pyruvate oxidation by pyruvate dehydrogenase (PDH) generates acetyl-CoA and NADH inside mitochondria as FAO. However, FAO induces a larger and more efficient increase in NADH, acetyl-CoA, and ATP levels per molecule of nutrient than PDH activity (i.e., glucose generates 2 acetyl-CoA, 1 C16-Fatty acid generates 8 acetyl-CoA). Moreover, the acetyl-CoA generated by FAO can enter the TCA cycle, to further increase NADH, FADH_2_, and ATP levels inside mitochondria. The larger increase in acetyl-CoA/CoA, NADH/NAD+, and ATP/ADP ratios during FAO results in a product-mediated decrease of PDH and different TCA cycle dehydrogenase activities (Figure 1A) [3]. This product-mediated inhibition, together with acetyl-CoA fueling citrate synthesis, causes a net increase in mitochondrial citrate levels that leads to its export to the cytosol. High cytosolic levels of citrate inhibit glycolytic enzymes preceding pyruvate synthesis, namely phosphofructokinase 1 (PFK1), PFK2, and pyruvate kinase (PK) (Figure 1A) [2]. Remarkably, PFK1 is the enzyme that commits glucose carbons to pyruvate synthesis, which means that PFK1 inhibition can selectively block the production of lactate or acetyl-CoA from glucose.

McGarry and colleagues defined the mechanism by which elevated glycolysis blocked mitochondrial FAO [4] (Figure 1B). Glycolysis increased malonyl-CoA synthesis in the cytosol, a direct inhibitor of mitochondrial carnitine-palmitoyl transferase protein 1 (CPT1) activity. CPT1 is located in the outer mitochondrial membrane and is responsible for the effective entry of long-chain FA from the cytosol into mitochondria. CPT1 is the bottleneck step determining FAO rates.

The elevation of malonyl-CoA synthesis as a result of increased glycolysis occurs as follows: glycolysis provides pyruvate to mitochondrial PDH, upregulating acetyl-CoA and NADH levels inside the mitochondria. Remarkably, NADH and acetyl-CoA are the exact same products generated by FAO. However, as PDH generates lower quantities of mitochondrial acetyl-CoA and NADH, it cannot downregulate TCA flux as FAO does (Figure 1B) [3]. In other words, PDH cannot consume mitochondrial CoA and NAD+ fast enough to decrease the activity of TCA cycle dehydrogenases that need these co-factors [3]. Moreover, the presence of pyruvate carboxylase (PC) activity inside the mitochondria allows pyruvate to be self-sufficient in preserving the total carbon pool of TCA cycle intermediates: PC directly transforms pyruvate into the TCA cycle intermediate oxaloacetate (OAA) (Figure 1B). Then citrate synthase can condense this PC-derived OAA with PDH-derived acetyl-CoA and produce citrate. As a result, elevated pyruvate availability in the mitochondria can generate more citrate than needed to sustain the TCA cycle carbon pool, causing citrate to be exported to the cytosol by the mitochondrial carrier SLC25A1/CIC. As FA can only generate acetyl-CoA in the matrix, FA cannot be self-sufficient as pyruvate to preserve the TCA cycle carbon pool. However, elevated citrate export is the same outcome after increasing mitochondrial FAO. Thus, how citrate export induced by glycolysis results in FAO inhibition? In the cytosol, glucose-derived citrate is broken down by ATP-citrate lyase (ACLY) to acetyl-CoA and OAA. Then, acetyl-CoA is carboxylated to generate the FAO-inhibitor malonyl-CoA, a reaction executed by acetyl-CoA carboxylases 1 and 2 (ACC1 and ACC2) that are encoded by different genes [5] (Figure 1B). Consequently, ACLY and ACC catalyze the essential steps that commit citrate export to FAO inhibition.

Remarkably, malonyl-CoA not only blocks mitochondrial FAO. Malonyl-CoA is the carbon precursor of de novo synthesized fatty acids in lipogenic tissues, such as liver and differentiating white adipocytes [5]. This dual role of malonyl-CoA has been used to justify the existence of two ACC isoforms: ACC1 commits malonyl-CoA to de novo fatty acid synthesis catalyzed by fatty acid synthase (FASN), while ACC2 commits malonyl-CoA to block FAO (Figure 1B) [5]. Supporting this conclusion, lipogenic tissues mostly express ACC1 and oxidative tissues that are unable to synthesize FA mostly express ACC2 (i.e., muscle). Interestingly, liver can execute both FAO and FA synthesis, explaining why similar amounts of ACC1 and ACC2 are expressed in hepatocytes [5]. Moreover, the association of ACC2 to the outer mitochondrial membrane allows generating malonyl-CoA in close proximity to its target, CPT1 (located in the outer membrane as well). On the other hand, cytosolic ACC1 would direct malonyl-CoA to cytosolic fatty acid synthase (FASN), not to mitochondrial CPT1 (Figure 1B) [5]. Another important aspect is that three different isoforms of CPT1 exist: CPT1-A, CPT1-B, and CPT1-C [6]. CPT1-C is a brain- and cancer-specific isoform with an unclear role in FAO. The most important functional difference between CPT1-A and B is that CPT1-A is less sensitive to malonyl-CoA-mediated inhibition than CPT1-B (IC_50_ = 116 vs. 0.29 µM). Finally, malonyl-CoA decarboxylase (MCD) is an enzyme that degrades malonyl-CoA and could relieve mitochondria from FAO blockage.

The identification and characterization of the enzymes determining the Randle cycle revealed that the following regulatory processes are the key determinants of the competition between FAO and glucose oxidation. However, there are still gaps in knowledge in these processes that need to be covered:(1)The mechanisms regulating ACLY, ACC1, and ACC2 activity and their subcellular distribution. Particularly: (a) How ACC2 association to the mitochondria is regulated; (b) whether ACLY could be located in close apposition to the mitochondria; (c) whether ACC1 can be kept far from mitochondria; and (d) how increased FAO blocks the activity of ACLY, ACC1, and ACC2 even in the presence of high glucose and independently of transcriptional changes. One can hypothesize that if citrate is not broken down close to ACC2 and mitochondria, this citrate might be preferentially routed toward FA synthesis.(2)The mechanisms determining FAO sensitivity to malonyl-CoA-mediated inhibition. A major process determining this sensitivity could be the regulation of CPT1-A and CPT1-B protein ratio in each tissue. If malonyl-CoA does not reach concentrations in the 100 µM range (IC_50_ for CPT1-A) in a CPT1-A-expressing tissue, FAO could still proceed despite elevated glycolysis.(3)The activity and subcellular distribution of malonyl-CoA decarboxylase (MCD), particularly the potential regulation of MCD apposition to mitochondria. The enzymatic degradation of malonyl-CoA close to the mitochondrial outer membrane could explain how some tissues might concurrently execute FAO and FA synthesis.(4)A major mechanism by which glucose restriction activates FAO and decreases malonyl-CoA synthesis is through an inhibitory phosphorylation of ACC1 and 2 executed by AMPK, a kinase activated when ATP levels drop [7]. However, Randle and colleagues observed that high FA availability could increase FAO in hearts perfused with glucose without changing ATP levels [1]. Therefore, it is a possibility that FA entry into the cells or its oxidation can decrease ACLY and ACC activities by a still uncharacterized post-translational mechanism. Other possibilities could include that FA activated AMPK by other mechanisms not involving decreased ATP content (i.e., ROS production derived from FAO) or that high levels of FA outcompeted the malonyl-CoA produced in tissues expressing CPT1-A.

A proper shift from anabolism and catabolism, and consequently the proper execution of the Randle Cycle, are not only essential processes to properly respond to nutrient availability. This shift from anabolism to catabolism is essential for cancer cells to survive metastasis or changes in their microenvironment. Thus, covering the gap in knowledge in the aforementioned mechanisms determining the Randle cycle raise the possibility of identifying new metabolic liabilities to stop cancer proliferation and survival. In this regard, these liabilities could be cancer-specific in tumors that escape the Randle cycle, as this escape could be considered a rare characteristic, even for cancer cells (Table 1). The following chapters will review the current knowledge of the proteins involved in the Randle cycle and their role described in colorectal cancers and lymphomas. We will describe some speculative theories on how cancers requiring both FAO and high glycolysis to proliferate specifically remodel their mitochondria to escape the Randle cycle, based on recently published mechanisms of FA synthesis and FAO regulation in healthy tissues. Finally, we will discuss how understanding the competition between glycolysis and FAO might reveal novel therapeutic approaches in cancer, as well as providing molecular mechanisms for dietary interventions on cancer.

## 2. The Role of Upregulated Glucose and Aminoacids Oxidation to Support Anabolism in Tumor Proliferation: It Is Not about ATP, but Anaplerosis and FA Synthesis

Tumor proliferation imposes a high demand for nutrients to fulfill the need to synthesize new proteins, lipids, and nucleic acids; as well as for the ATP required to energize these biosynthetic processes. The upregulation of glucose oxidation to lactate in tumors is a major metabolic program that sustains uncontrolled growth rates, as shown by Warburg [8,9,10]. Initially, it was proposed that the Warburg effect was an indirect consequence of proliferation, namely an adaptive response to the exceedingly high ATP demand imposed by the augmented rates of cell division. However, the relatively recent demonstration that the transcription of genes encoding glycolytic enzymes is directly upregulated by the action of oncogenes, by the loss of genes encoding tumor suppressors and by kinases that also signal proliferation, support that higher glycolytic rates in tumors are not simply a secondary or adaptive response to elevated ATP demand (or defined by others in the literature as a “passive response”). Recent studies demonstrate that glycolytic intermediates and precursors of pyruvate, such as 3-phosphoglycerialdehide, are needed as a carbon source to fuel the synthesis of amino and nucleic acids at the high rates imposed by tumor proliferation [11,12]. Moreover, glucose also fuels the pentose phosphate pathway [13], which is needed to synthesize nucleotides, generate NADPH for lipid biosynthesis, and to counteract elevated ROS production caused by the loss of tumor suppressors or by stressors of the tumor environment [14,15]. In addition, not all pyruvate generated by glycolysis is used to produce lactate. A fraction of glucose-derived pyruvate is diverted to mitochondria to produce citrate and fuel FA and lipid biosynthesis, with about 60% of fatty acyl carbons are derived from glucose in glioblastoma [16]. Glioblastoma also consumes the amino acid glutamine to replenish TCA intermediates needed for fatty acid (FA) synthesis, probably to compensate for the pyruvate being consumed to generate lactate and thus diverted away from TCA anaplerosis [16,17,18,19]. Consequently, amino acids play a role in cancer beyond ammonia provision, protein and nucleotide synthesis.

Accordingly, tumors are not just highly dependent on glucose, but also on exogenous glutamine to proliferate and survive [20]. The addiction of colorectal cancer to glutamine can be considered as strong as the addiction to glucose. Indeed, glutamine is sufficient to sustain cell growth in glucose-deprived colorectal cancer cells, while fatty acids are unable to do so [21]. This inability of fatty acids to preserve proliferation shows that nutrient availability in colorectal cancer might regulate anabolism and catabolism as in healthy tissues: high extracellular availability of glucose and amino acids allows high glycolytic and TCA fluxes to fuel ATP, fatty acid synthesis, and other anabolic processes, whereas FAO by itself cannot efficiently support anabolism and is mostly recruited to sustain ATP and NADPH production needed to survive nutrient scarcity (Table 1).

## 3. The Role of Mitochondrial Fatty Acid Oxidation (FAO) Supporting Cancer Survival and Proliferation

The preference for glucose over fatty acids to sustain cancer cell proliferation supports that the need for efficient ATP provision is not the main driver for glucose preference in cancer. FAO is more efficient generating ATP per molecule of nutrient than glucose-derived pyruvate oxidation. Moreover, more recent studies demonstrate that FAO can provide ATP as fast as required to support cancer proliferation [22]. Accordingly, FAO can even support anabolism in Warburg-abiding colorectal cancer cells, but only as an adaptative response to extracellular acidification [22]. This increase in FAO was even preserved when these acidosis-adapted cancer cells generated in vitro were grown as tumor xenographs in mice [22]. High glycolytic rates can acidify the cytosol and the extracellular space via the generation of lactate, meaning that accumulation of acid can cause product-mediated inhibition of glycolysis. As a result, this shift to FAO is thought to be a response to an acidosis-induced decrease in glycolysis to rates that cannot sustain proliferation.

In this regard, Corbett et al. observed that acidosis caused a decrease in glucose consumption and pyruvate oxidation in the mitochondria, pointing to a reduction in pyruvate synthesis by glycolysis as well [22]. Remarkably, acidosis resulted in increased glutamine oxidation, providing alpha-ketoglutarate to preserve TCA cycle anaplerosis and thus proliferation. In other words, alpha-ketoglutarate stemming from glutamine oxidation allowed replacing glucose-derived acetyl-CoA and OAA as carbon sources to synthesize citrate and fatty acids. In addition, glutaminolysis generates ammonia that could counteract acidification and prevent the other toxic effects associated with low pH. Remarkably, Corbett et al. also identified that the increase in acetyl-CoA caused by FAO changed histone acetylation in the nucleus, which selectively decreases the transcription of ACC2, but not of ACC1. The selective decrease in ACC2 provided a mechanism by which these cells could concurrently perform FAO and FA synthesis under stress (see Figure 1B). In all, this study could be considered as the exception that proves the rule that FAO is used for catabolism and glycolysis for anabolism in colorectal cancer (Table 1) [22].

If FAO can support high proliferation rates in cancer, why colorectal cancers choose glycolysis instead of FAO combined with glutamine oxidation to support proliferation under non-stressed conditions? The reason might be that FAO does not provide a selective advantage and introduces a metabolic liability that decreases cancer cell aggressiveness. Supporting this conclusion, the provision of palmitoyl-carnitine, a form of fatty acids that directly enters into mitochondria, elevated H_2_O_2_ production to rates that caused cell death of colorectal cancer cells. The increase in H_2_O_2_ production was likely caused by the increase in the mitochondrial NADH/NAD+ ratio, also known as reductive stress [23]. As a consequence, it seems as nutrients or metabolic strategies inducing reductive stress in the mitochondria have not been selected for cancer cell proliferation. These additional findings on FAO in colorectal cancer cells further support the idea that NADH and ATP synthesis capacity and efficiency are not the main parameters driving the metabolic rewiring observed in cancer.

Despite of most cancers elevate glycolysis to support fatty acid (FA) synthesis and proliferation in non-stressed conditions [24], a subset of B-cell lymphomas use an upregulation of mitochondrial FAO as a primary metabolic strategy to support anabolism and proliferation (Table 1) [25,26]. Furthermore, the upregulation of FAO in these tumors is concurrent with elevated glycolysis and FA synthesis. These studies concluded that FAO was recruited to meet the uniquely high ATP demand [25,26], stemming from this subset of lymphomas growing much faster than other solid tumors [25,26]. In addition, FAO provided reducing equivalents to mitochondria, which are needed for antioxidant glutathione to effectively counteract increased ROS production caused by their growth in suspension [25].

**Table 1 cells-09-02600-t001:** Nutrient preference reported in Randle cycle abiding and non-abiding tumors and their response to environmental stress and metastasis.

	**Tumor Type**	**Primary (Warburg Effect)**	**Metastasis/Environmental Stress**	**Reference**
**Randle Cycle**	Colorectal cancer	High glycolysis to lactate Dependent on glucose and glutamine to proliferate. Glucose and glutamine used for FA synthesis.	FAO is elevated as an acidic stress response. FAO is needed to survive detachment and the metastatic process.	[21,22,23,27]
	Glioblastoma	High glycolysis to lactate Dependent on glucose and glutamine to proliferate. Glucose and glutamine used for FA synthesis.	FAO is recruited to survive stresses imposed by the microenvironment. FAO is recruited to protect from oxidative stress (elevated ROS).	[16,28,29,30]
	**Tumor Type**	**Primary (Non-Solid Tumors)**	**Metastasis/Environmental Stress**	**Reference**
**Non-Randle Cycle**	Lymphomas/ Leukemias	High glycolysis to lactate Dependent on glucose and glutamine to proliferate. FAO needed to meet ATP demand and counteract high ROS production caused by growth in suspension.	The primary tumor already show a shift in nutrient preference similar to solid tumors when they are detaching and metastasizing. FAO is not a defining feature of all non-solid tumors, as a subset of B-cell lymphoma are Warburg-like and do not recruit FAO.	[25,26,31]

With the existence of the Randle cycle and other competitive mechanisms that regulate nutrient utilization independently of hormones and the ATP/ADP ratio, the fact that FAO can support proliferation in highly glycolytic cancers leaves some unanswered questions:(1)What is the main molecular mechanism allowing FAO and FA synthesis to concurrently occur in this subset of B-cell lymphomas? Are the epigenetic changes mediated by acetyl-CoA decreasing ACC2 transcription responsible [22] or other post-translational mechanisms exist?(2)Is there a reason beyond the higher demand for ATP and NAD(P)H synthesis associated with being a non-solid tumor to choose FAO, as suggested by the high proliferation rates of the other subset of B-cell lymphomas that do not use FAO?(3)Is there any change in the glycolytic machinery of tumors that could make them insensitive to the inhibitory effects of FAO on glycolysis? On the contrary, is there a change in the mitochondrial machinery executing FAO to make them insensitive to glycolysis-mediated inhibition of FAO?

In the following subchapters, we will review the studies that can answer these questions, as well as speculate how FAO and FA synthesis concurrently occur in these cancers escaping the Randle cycle.

### 3.1. Glycolytic Reprogramming in Colorectal Cancers Allows a Randle Cycle-Abiding Behavior and Recruit FAO to Survive Redox Stress and Metastasis

Instead of adult tissue isoforms of pyruvate kinase (PKM1, PKL or PKR), cancer cells express PKM2, the embryonic and stem cell isoform of PK. PKM2 increases glycolic flux and decreases the dependency on mitochondrial oxidative phosphorylation to proliferate [32]. Moreover, the reaction transforming phosphoenolpyruvate (PEP) to pyruvate catalyzed by PKM2 is not regulated by ATP levels, decoupling glycolytic flux from ATP demand [11,33]. This decoupling allows diverting glycolytic intermediates upstream of PEP toward anabolic reactions independently of the ATP/ADP ratio. Moreover, PKM2 activity preserves its sensitivity to glucose levels and is activated by the glycolytic intermediate fructose 1,6-bisphosphate (F-1,6-BP). These properties of PKM2 enzymatic activity highlight that the major driver for glycolysis remodeling in cancer might be increasing anabolic capacity, rather than ATP synthesis.

Interestingly, colorectal cancer cells (CRC) express PKM2. Counterintuitively though, despite F-1,6-BP activates PKM2, high F-1,6-BP concentrations can decrease CRC proliferation [34]. This bell shaped relationship between F-1,6-BP levels and CRC proliferation could explain how some glycolytic tumors could benefit from FAO. A decrease in PFK1 activity caused by FAO-derived citrate would prevent an increase in F-1,6-BP to levels that block proliferation. Alternatively, FA synthesis in CRC might have a high demand for citrate, with high citrate consumption toward malonyl-CoA keeping citrate to concentrations that cannot block PFK1 [35].

Another aspect is that not all glucose that supports tumor proliferation is transformed to pyruvate. Glucose enters the pentose phosphate pathway (PPP), which provides carbons for nucleotide synthesis and electrons for the NADPH needed to eliminate ROS. Tumors show increased ROS production as a result of the loss in tumor-suppressor gene function, environmental stresses, and inflammation. Thus, the inhibition of PFK1 mediated by FAO-derived citrate could be helping to increase glucose availability for PPP. Supporting this conclusion, glucose restriction in lung cancer cells activated AMPK to increase FAO [36]. The main goal of AMPK activation and increased FAO in these glucose-restricted tumors was to inhibit FA synthesis and generate NAD(P)H, to preserve antioxidant capacity, rather than preserving the ATP/ADP ratio [36]. In this regard, it is a possibility that FAO induced by AMPK, by inhibiting PFK1, could contribute to NADPH generation by increasing glucose availability for the PPP. Remarkably, AMPK activation blocks both ACC1 and ACC2, enzymes needed for de novo lipogenesis from glucose. Consequently, the increase in glycolytic flux and FAO observed in healthy tissues when AMPK is activated [7] would be incompatible with FA synthesis from glucose, essential for some tumors to proliferate. Interestingly though, blocking only one isoform of ACC (ACC1 or ACC2) could increase lung cancer proliferation [36]. However, it is likely that a double KO of ACC1 and ACC2 would be lethal for these tumors as well, which would recapitulate better the effects of AMPK activation on de novo lipogenesis.

Altogether, these studies support that the reprogramming of glycolysis in cancer can abide to the Randle cycle. Furthermore, they show that FAO is recruited in colorectal cancer as an adaptation to survive stressful environments or glucose restriction, when anabolism can be deleterious for tumor survival. A major detrimental factor of anabolism is that it limits ROS detoxification, with increased ROS to damaging levels being a common stress induced by cell detachment, loss of tumor suppressor function, and metastasis [36,37,38]. 

Further supporting the preferential recruitment of FAO under stress, FAO can be activated in tumor cells by other extracellular signals as well, and not only by decreased glucose availability or elevated intracellular ROS production. Cyclooxygenase-2-derived prostaglandin E2 (PGE2) activates the transcription factor NR4A and induces the expression of FAO enzymes in colorectal cancer [39]. PGE2 is increased by inflammation, meaning that colorectal cancer cells can sense inflammation and respond by increasing mitochondrial FAO to survive. In colorectal cancer cells under nutritional and environmental stress, circACC1 levels are increased to regulate lipid metabolism. circACC1 is a non-coding RNA splicing generated from ACC1 RNA. circACC1 was shown to activate AMPK, which allowed preserving the ATP/ADP ratio by increasing FAO and glycolysis in cancers under environmental stress [40]. Therefore, this is another example that specific signaling mechanisms exist to sense stress and increase FAO as a survival response, even by a circRNA generated from the transcript that encodes a lipogenic and anti-FAO gene (AAC1). 

### 3.2. Does Metastasis Mimic the Fasting State as the Randle Cycle Defined, Explaining the Shift to FAO?

During fasting, fatty acids released from the adipose tissue are oxidized by muscle mitochondria, which inhibits glycolysis and PFK activity. Metastasis is considered a state of nutrient deprivation and of increased ROS production to damaging levels. Thus, one can conclude that the metastatic process of a tumor could mimic the change in nutrient preference in muscle during fasting, as defined by the Randle cycle.

During metastasis, total mitochondrial mass increases concomitantly with OXPHOS dependency, which fits an upregulation of FAO [41,42,43,44]. Alexidine, a drug blocking cardiolipin synthesis in mitochondria, and concomitantly OXPHOS, decreases invasiveness of lung cancer [42]. Metabolomic and transcriptomic analysis of tumor-draining lymph nodes revealed that metastatic tumors show an upregulation of mitochondrial FAO, with FAO inhibition blunting tumor invasiveness [45]. In ovarian and colorectal cancer, adipocytes play an important role in metastasis by supplementing cancer cells with fatty acids [46,47]. Metastatic cells elevate fatty acid binding protein 4 (FABP4) expression and secretion, which favors the uptake of fatty acid released from adipocytes to cancer cells and allows FAO to fuel proliferation [47]. Consequently, these studies demonstrate that increasing fatty acid availability and FAO are vital for the success of different cancer metastasis and survival in the microenvironment. Maybe as the exception that proves the rule, when metastatic ovarian cancer cells detach (anoikis), pyruvate is the substrate that increases invasiveness and supports mitochondrial ATP production [44]. Thus, it seems as FAO is not unequivocally associated with metastasis and some tumors might use glucose-derived pyruvate to meet the high demand in ATP and counteract oxidative stress.

FAO was shown to be a key factor in colorectal cancer metastasis, as well as for survival in a stressful tumor microenvironment. Adipocytes were shown to contribute to colorectal tumor growth by providing fatty acids to fuel FAO in the tumor [46]. CPT1-A-mediated FAO prevents anoikis in colorectal cancer cells, which is essential for survival during metastasis. Furthermore, the metastatic tumors generated from colorectal cancer showed increased levels of CPT1-A expression [48]. Thus, one can conclude that the Randle cycle is preserved in colorectal cancers, with the shift to FAO being a survival response to glucose limitation, increased oxidative stress, and/or inflammation.

Remarkably, colorectal cancer cells not only express CPT1-A, they also express CPT1-C. Silencing of mitochondrial CPT1-C in unstressed colorectal cancer causes intracellular fatty acid accumulation, lipid peroxidation, and decreased ATP levels [49]. Thus, FAO was concluded to be a protective mechanism to license cancer cell proliferation by preventing lipotoxicity, even in unstressed colorectal cancer. Indeed, CPT1-C is the cancer- and brain-specific isoform of mitochondrial CPT1 that is inhibited by malonyl-CoA [6]. However, there is conflicting evidence on the role of CPT1-C in FAO and its inhibition by malonyl-CoA in cancer. CPT1-C is unable to add carnitine to fatty acyl-CoA moieties, meaning that CPT1-C cannot promote FA entry into the mitochondria and thus FAO. As a result, the cellular dysfunction induced by CPT1-C downregulation might not be caused by decreased FAO.

### 3.3. How Do B-Cell Lymphomas and Leukemias Concurrently Perform FAO and FA Synthesis, Escaping from the Randle Cycle?

The subset of B-cell lymphomas and leukemias that use FAO to proliferate express CPT1-A: the isoform with lower sensitivity to malonyl-CoA [25] and found in Randle-cycle abiding and lipogenic tissues, such as liver [6]. As it is only a subset of lymphomas that shows FAO preference, it seems as this preference for FAO to support proliferation is genetically determined, rather than being a response to a challenging environment. However, it is unclear how FAO-dependent B-cell lymphomas show a preferential expression of CPT1-A and whether some post-translational mechanisms could explain FAO preference.

Prolyl hydroxylase 3 (PHD3) is an enzyme that senses the energetic status of the cell and can hydroxylate different proteins. PHD3 hydroxylates ACC2, which increases its acetyl-CoA carboxylase activity and results in elevated malonyl-CoA synthesis, consequently blocking FAO [50]. Acute myeloid leukemia (AML) expresses low levels of PHD3, which was associated with their dependency on FAO to proliferate [31]. It is possible that in low PHD3-expressing cancers, CPT1-A will be more active. The reason is that with ACC2 downregulation, most malonyl-CoA will be generated far away from mitochondrial CPT1, preventing malonyl-CoA action blocking FAO. Thus, PHD3-mediated hydroxylation of ACC2 constitutes one of the few post-translational mechanism that determine FAO and FA synthesis concurrency in leukemias, independently of changes in ACC2 transcription by epigenetic modifications. However, low PHD3 expression cannot explain why elevated FAO and citrate accumulation (expected from absent ACC2) are not blocking PFK1 in AML, allowing glycolysis to operate at very high rates. Moreover, how low levels of PHD3 expression are selected in AML tumors is still unclear. Remarkably though, cancer cells with high PHD3 expression are less reliant on FAO, while silencing PHD3 stops their proliferation [31].

### 3.4. Can Mitochondrial Heterogeneity and Dynamics Explain How B-Cell Lymphoma and Leukemias Concurrently Perform FAO and FA Synthesis?

The study identifying this FAO-dependent subset of B-cell lymphomas proposed that high rates of FA synthesis impeded that enough malonyl-CoA accumulated to block FAO [25]. Supporting this premise, this subset of B-cell lymphomas expressed high levels of CPT1-A, the isoform needing higher malonyl-CoA concentrations to be inhibited. However, the role and regulation of ACLY, ACC1, and ACC2 in these FAO-dependent B-cell lymphomas remained unaddressed. The more recent findings of different populations of mitochondria existing inside the same cell support an alternative explanation for the concurrency of FAO, FA synthesis and glycolysis. This alternative could be independent of PHD3 activity and/or of acetyl-CoA-mediated epigenetic remodeling to selectively decrease ACC2 transcription.

The major process responsible for segregating mitochondrial populations inside the same cell is mitochondrial dynamics, namely mitochondrial motility fusion and fission events. In this regard, changes in extracellular nutrient availability can modulate mitochondrial dynamics, independently of changes in hormonal action [51]. An excessive provision of FA over high glucose caused mitochondria inside the same cell to be functionally heterogeneous, a process that also occurs when mitochondrial fusion is blocked using genetic approaches [52,53]. In addition, we demonstrated that a segregated population of mitochondria is specialized to incorporate FA into lipid droplets, which we named peridroplet mitochondria [54]. With these characteristics of mitochondria in healthy tissues, we speculate that the following mechanisms could explain Randle-cycle avoidance in cancer cells.

We propose that two segregated mitochondrial populations exist in cancer, lipogenic mitochondria, and cytosolic mitochondria (Figure 2). This segregated population of lipogenic mitochondria would be qualitatively equivalent to the peridroplet mitochondria that we published in BAT. Peridroplet mitochondria had a higher capacity to oxidize pyruvate, higher citrate synthase activity, and decreased capacity to execute FAO [54]. We propose that lipogenic mitochondria in cancer will preferentially associate with ACC2 and express CPT1-B. The combination of ACC2 and CPT1-B would allow glycolysis to effectively block FAO in lipogenic mitochondria, impeding the decreases in TCA and PDH activity induced by FAO. Then, the population of cytosolic mitochondria would be the one executing FAO, as cytosolic mitochondria would carry CPT1-A and would not associate with ACC2. Consequently, cytosolic mitochondria could not be exposed to inhibitory malonyl-CoA concentrations, as CPT1-A has an IC50 of 100 μM. These cytosolic mitochondria could associate with malonyl-CoA decarboxylase (MCD) when the glycolytic and lipogenic fluxes were high enough to generate malonyl-CoA at concentrations > 100 μM. An unaddressed aspect of this model is how the increase in citrate production by both cytosolic mitochondria and lipogenic-mitochondria is not blocking PFK1 activity. Another speculative hypothesis would be that ACLY could transiently associate with the outer mitochondrial membrane of lipogenic mitochondria (or even cytosolic mitochondria). In this way, citrate could be rapidly transformed to acetyl-CoA and OAA before having the opportunity to inhibit PFK1 and glycolysis. To our knowledge, ACLY has not been found to be associated with mitochondria. Thus, this is raising the exciting possibility that ACLY association to mitochondria is a specific requirement of non-Randle cycle abiding tumors.

## 4. Evidence That Targeting Mitochondrial-Derived FA and FAO Synthesis Is an Anti-Tumor Strategy

### 4.1. Examples of Blocking FAO as a Broad Anti-Cancer Strategy

The degree of tumor malignancy is positively correlated with FAO in multiple cancers, not only in lymphoma and colorectal cancer [55,56,57]. For instance, CPT1-A expression has been used as a prognosis marker for AML and triple-negative breast cancer (TBNC) severity. AML patients with high CPT1-A expression show lower survival [55]. TNBC, a highly metastatic tumor, expresses the protein CUB-domain containing protein 1 (CDCP1), which activates acyl-CoA synthesis to increase FAO [56]. Elevated FAO in TNBC decreases the size and numbers of lipid droplets (LD), with low LD content in TBNC being associated with higher invasion capacity [56]. Conversely, preserving LD content decreases TNBC malignancy and metastases. These studies suggest that blocking FAO could be an approach to prevent TNBC metastasis and stop AML progression.

In glioblastomas, blocking FAO with etomoxir, a panCPT-1 inhibitor, was sufficient to activate ROS-induced cell death [30]. Likewise, etomoxir treatment of bladder cancer cells caused cycle arrest via PPAR-γ activation and blunting Akt phosphorylation [58]. Other approaches targeting other components of FAO different than CPT-1 stopped tumor growth as well [49,59,60,61].

Another anti-cancer strategy induces dependency on FAO and then specifically blocks FAO or even mitochondrial OXPHOS. Inhibition of cyclin-dependent kinase 9 (CDK9), an RNA-polymerase II regulator, activates AMPK to increase FAO in prostate cancer cells [62]. The combination of CDK9 and FAO inhibition stopped cell proliferation efficiently [62]. Furthermore, new compounds targeting the receptor kinase c-MET increased mitochondrial FAO and OXPHOS in glioblastoma, inducing sensitivity of tumor xenographs in mice to FAO-specific inhibitors [63].

Ovarian cancer patients with low expression of the tumor suppressor NKX2–8 have a poor prognosis, with the deletion of NKX2-8 activating FAO and increasing chemoresistance [64]. NKX2-8 (Homeobox protein Nkx-2.8) is a transcription factor regulating development. The loss of NKX2-8 dissembles a deacetylation complex, resulting in chromatin acetylation leading to increased CPT1-A and CPT2 expression in ovarian cancer [64]. Adipocytes were shown to provide a favorable environment for the growth of these NKX2-8 deleted ovarian tumors, as adipocytes retained chemotherapeutic reagents and provided fatty acids to fuel FAO [64]. Thus, blocking FAO or even decreasing fatty acid availability could be a strategy to stop ovarian tumor growth.

### 4.2. Examples of Blocking Lipid Synthesis as a Potential Therapeutic Strategy

Treatment of lymphomas with Orlistat, which is a lipase and fatty acid synthase (FASN) inhibitor, stopped tumor growth in vivo [65]. Orlistat treatment limited the availability of fatty acids required to produce biomass and counteract the increase in ROS production. The latter was explained by the need of de novo lipogenesis to produce endogenous antioxidants. Likewise, drug screening approaches provided new FASN inhibitors that efficiently stopped colorectal cancer growth [66]. Interestingly, disrupting FASN activity induced apoptosis through malonyl-CoA accumulation [67]. Thus, it is possible that citrate accumulated as well as a result of FASN inhibition, which could further contribute to cell death by blocking glycolysis.

### 4.3. Examples of Stimulating FAO as a Mechanism to Stop Cancer Growth

The Randle cycle shows that FAO stimulation can block glycolysis. Given the importance of the Warburg effect in glucose-dependent tumors, one could hypothesize that a less toxic approach to block glycolysis is to stimulate FAO in Warburg-like tumors.

Changing mitochondrial function by overexpressing UCP3 increased FAO independently of ATP demand, which also decreased Akt phosphorylation [68]. Rescue of pAkt restored proliferation in UCP3 overexpressing cancer cells. This Akt-mediated rescue led to the conclusion that signals derived from FAO inhibiting Akt, rather than FAO itself, were responsible for stopping cell proliferation. In colorectal cancer, HIF1a inhibits FAO to promote cell proliferation. Accordingly, Oroxylin A, which reduces HIF1a expression and activates FAO, induced cell cycle arrest [69].

## 5. Could Forcing the Mitochondria to Use an Inadequate Fuel Induce the Integrated Stress Response (ISR) to Stop Tumor Growth?

The approaches discussed in Section 4 change mitochondrial ATP synthesis or OXPHOS, either by suppressing or activating FAO. The antineoplastic approaches that target metabolism aim to selectively disrupt cancer-specific energetic and anabolic processes, leading to an activation of stress responses that stop cell growth and even cause cell death. In this regard, the stress induced to mitochondria by manipulating extracellular nutrient availability or mitochondrial fuel preference might be an approach with therapeutic relevance. In a similar way that exposure to high levels of fatty acids was sufficient to stop glycolysis in muscle by inducing mitochondrial FAO [1], one could imagine that changing nutrient availability can generate new metabolic liabilities in tumors by modulating mitochondrial function. One of the stress pathways activated by a change in mitochondrial OXPHOS is the integrated stress response (ISR) [70,71,72,73,74,75,76,77].

Independent studies showed that the heme-regulated eIF2α kinase (HRI) is the main kinase responsible for sensing a disruption in mitochondrial OXPHOS to activate the ISR. However, the molecular mechanism by which a disruption in OXPHOS activated HRI was only recently identified. Mitochondrial depolarization, decreased mitochondrial ATP levels and elevated mitochondrial ROS can activate the mitochondrial inner-membrane protease OMA1 [78]. Remarkably, all these mitochondrial parameters can be modulated by extracellular nutrient availability [51]. Thus, OMA1 is considered a stress-responsive protease that senses a decline in mitochondrial ATP synthesis as a result of mitochondrial dysfunction. The key finding connecting OXPHOS to HRI activation was the demonstration that OMA1 cleaved the mitochondrial protein DELE1 [78]. The cleaved form of DELE1 can be released to the cytosol, where it directly binds to HRI and activates its function (Figure 3) [78]. HRI phosphorylates the eukaryotic translation initiation factor eIF2a, which inhibits protein translation [79], with the exception of ATF4 protein. ATF4 is a transcription factor translocating from the cytosol to the nucleus, where it regulates the expression of genes controlling cell cycle, represses the expression of mitochondrial nuclear-encoded genes and it can even activate a cell death program [70,71,72,73,74,75,76,77]. Thus, the ATF4-mediated transcriptional response is the executor of the ISR. ATF4 activity can lead to either survival, at the expense of shutting down mitochondrial function, or to cell death, depending on the cell and intensity of the stress. The new mechanistic connection between mitochondrial OXPHOS and ISR [78] raises the possibility that altering nutrient and fuel availability to mitochondria could activate DELE1, HRI, and ATF4 to stop tumor growth (Figure 3). There are studies already supporting this possibility.

In prostate cancer, blocking the mitochondrial pyruvate carrier (MPC) decreases the levels of citrate and other TCA intermediates, which limits lipid biosynthesis and leads to ISR activation. Also, MPC blockage increases glutamine consumption to compensate for the absence of mitochondrial pyruvate as carbon source to fuel anabolism and TCA anaplerosis [70]. We showed that MPC inhibition in brown adipocytes also increases glutamine utilization and elevates FAO [80]. Thus, it is indeed a possibility that the shift to FAO induced by MPC inhibition is responsible for ISR activation in prostate cancer, being sensed as a mitochondrial stress.

Another potential target to modulate mitochondrial pyruvate utilization is the mitochondrial protein FAM210B, as its downregulation increases pyruvate dehydrogenase (PDH) activity [81]. Breast cancer cells with high levels of FAM210B show reduced PDH activity and more invasiveness [81]. The decrease in mitochondrial pyruvate oxidation caused by FAM210B overexpression promotes tumor progression in breast cancer, which is the exact opposite to what was observed in prostate cancer [81]. This raises the possibility that only when FAO is activated, MPC inhibition results in ISR activation to stop tumor growth. In gastric cancer, the inhibition of mitochondrial ATP synthesis activates the ISR executor ATF4, but in this case ATF4 was a pro-survival response [75]. Indeed, ISR-ATF4 activation in gastric cancer leads to cisplatin and cell death resistance [75]. This latter study suggests that the activation of ISR-ATF4 response stops tumor growth only in OXPHOS-dependent cancers. On the other hand, the ISR-ATF4-mediated suppression of mitochondrial function can generate an additional advantage to Warburg-like tumors.

## 6. Conclusions and Future Perspectives

The metabolism of cancer is incredibly plastic, as mitochondria are. The hormone/signaling-independent regulation of mitochondrial function by nutrient availability, as revealed by the Randle cycle, can provide novel strategies to stop cancer cell growth. Thus, unravelling the molecular mechanisms that allow cancer cells to abide or escape the Randle cycle could be inspiring novel therapeutic strategies, particularly to kill tumors dependent on concurrent glycolysis and FAO. The identification of these molecular mechanisms might provide the molecular basis for dietary interventions to treat cancer as well. Such molecular mechanisms could involve the activation of OMA1 by changes in mitochondrial OXPHOS, to engage the HRI-ATF4 axis of the ISR to stop tumor growth and/or induce cell death.

## Figures and Tables

**Figure 1 cells-09-02600-f001:**
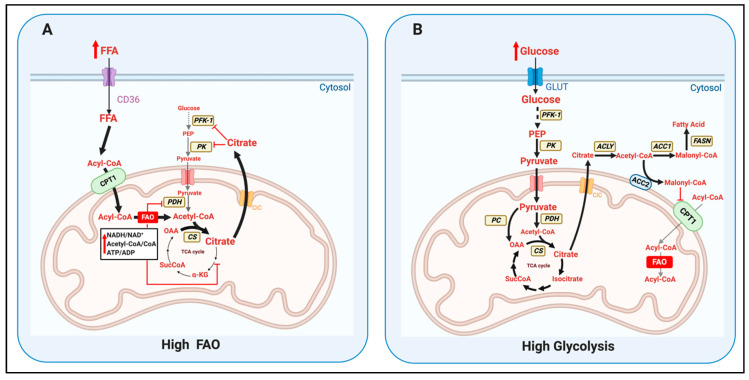
The key enzymes determining the competition of glycolysis versus mitochondrial fatty acid oxidation (FAO) defined by the Randle cycle. (**A**) Increased extracellular free fatty acids availability (FFA) can increase mitochondrial fatty acid oxidation (FAO), resulting in higher levels of acetyl-CoA, NADH, and ATP inside the mitochondria. As a result, citrate production by citrate synthase (CS) is increased, but the activity of PDH and TCA cycle dehydrogenases is decreased by the elevation in acetyl CoA/CoA, NADH/NAD+, and ATP/ADP ratios. Thus, FAO leads to an accumulation of mitochondrial citrate, which is exported to the cytosol via the CIC/SLC25A1 exporter. High citrate in the cytosol can inhibit phosphofructokinase-1 (PFK-1) and pyruvate kinase (PK) activities, decreasing glycolysis, pyruvate synthesis and oxidation. (**B**) When extracellular glucose increases, the concomitant upregulation in glycolysis provides more pyruvate to the mitochondria. Pyruvate oxidation by PDH and pyruvate carboxylation by pyruvate carboxylase (PC) generate acetyl-CoA and OAA respectively, increasing citrate synthase (CS) activity. The amount of citrate produced is higher that needed to sustain the carbon pool of TCA cycle intermediates, which causes its export to the cytosol. Under high glucose, cytosolic citrate is hydrolyzed by ATP-citrate lyase (ACLY) into acetyl-CoA and OAA. Acetyl-CoA is carboxylated by two acetyl-CoA carboxylases, ACC1 and ACC2, to generate malonyl-CoA. ACC2 is localized in the outer mitochondrial membrane, causing malonyl-CoA to be more accessible to carnitine-palmitoyl transferase 1 (CPT1). Malonyl-CoA inhibits CPT1 activity, the protein responsible for the entry of long chain fatty acids into mitochondria. ACC1 is cytosolic and can preferentially commit malonyl-CoA to the cytosolic enzyme fatty acid synthase (FASN), responsible for de novo fatty acid synthesis.

**Figure 2 cells-09-02600-f002:**
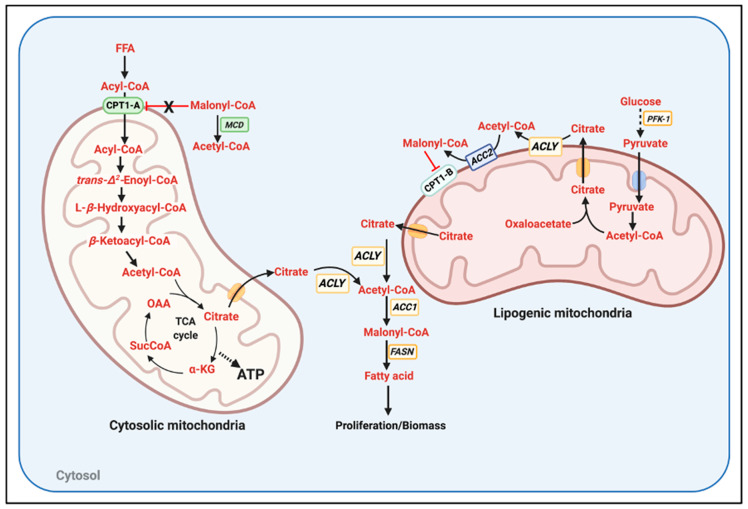
The potential existence of two segregated populations of mitochondria in cancer: lipogenic versus fatty acid oxidizing mitochondria. Two segregated populations of mitochondria inside the same cell can explain the concurrence of glycolysis and FAO in lymphoma and leukemias. We demonstrated the existence of these functionally segregated populations of mitochondria in brown adipose tissue [54]. In this model, cytosolic mitochondria consume free fatty acids for FAO, providing ATP, NADPH and likely other factors required for proliferation. Cytosolic mitochondria harbor CPT1-A but do not have ACC2 (Acetyl-CoA Carboxylase 2) associated, hindering malonyl-CoA-mediated inhibition of FAO. On the other hand, lipogenic mitochondria carry ACC2, CPT1-B, the isoform with higher sensitivity to malonyl-CoA inhibition and maybe have ACLY close (ATP-citrate lyase). In addition, lipogenic mitochondria would have a higher capacity to oxidize pyruvate and synthesize citrate, as we reported in brown adipose tissue, allowing a highly effective and local generation of malonyl-CoA. Lipogenic mitochondria could be associated to lipid droplets or to the ER. Moreover, if citrate generated by both cytosolic and lipogenic mitochondria generated an amount of malonyl-CoA that overcame fatty acid synthase (FASN) capacity, malonyl-CoA decarboxylase (MCD) could be recruited to cytosolic mitochondria. MCD would eliminate malonyl-CoA only in close vicinity to cytosolic mitochondria, preventing FAO blockage but allowing FASN to continue in the cytosol.

**Figure 3 cells-09-02600-f003:**
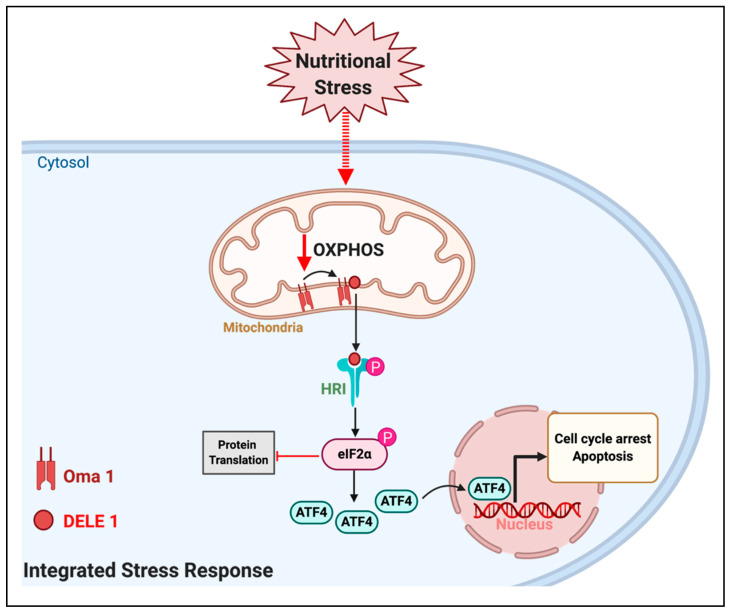
The effects of extracellular nutrients on mitochondria could activate the integrated stress response (ISR) pathway in cancer. Changes in extracellular nutrient availability can induce changes in mitochondrial function leading to decreased oxidative phosphorylation (OXPHOS), namely mitochondrial ATP synthesis fueled by respiration. A decrease in OXPHOS can activate the mitochondrial protease OMA1, which cleaves the mitochondrial protein DELE1. The short form of DELE 1 can be released to the cytosol, bind to the Heme-regulated inhibitor kinase (HRI) and activate its function. HRI phosphorylates the eukaryotic translation initiation factor 2a (eIF2 a), which inhibits protein translation, with the exception of the transcription factor ATF4. The actions of ATF4 in the nucleus execute the ISR to induce cell cycle arrest and even apoptosis.

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
