# Peer review of "The Role of Mitochondrial Fat Oxidation in Cancer Cell Proliferation and Survival"

_cells, 2020, doi:10.3390/cells9122600_

Round 1

Reviewer 1 Report

This is a well written review that comprehensively describes how mitochondrial fatty acid oxidation is modulated in cancer, and how this may provide a therapeutic target for cancer treatment. In particular, the review notes the differences in FAO upregulation (or downregulation) in different types of tumors, and also how this may affect metastatic potential. It also highlights where knowledge is still lacking in the field and suggests possible mechanisms that can be explored in the future.

I found the manuscript well organized and generally well written, with the different sections logically organized. Overall, I fully support acceptance of this manuscript, and only have some minor comments that require addressing.

1) Line 59/60: 'It is a possibility that decreased pyruvate utilization is directly driven by media acidification'. Are you talking about in vitro experiments here, and therefore the media would acidify? From the context of the paragraph I am assuming that your are talking about cancer cells in vivo; therefore a different term to 'media' would be more appropriate.

2) Please define the abbreviations in Figure 1 and Figure 2, notably FASN and PDM

3) Line 84: do you mean escape?

4) Please check spelling/grammar at Lines 96, 112, 114, 121/122, 124, 129, 151, 255, 274, 279, 307, 313, 315.

5) Line 234: do you mean 'form of' instead of 'from'?

Author Response

Reviewer # 1:

1) Line 59/60: 'It is a possibility that decreased pyruvate utilization is directly driven by media acidification'. Are you talking about in vitro experiments here, and therefore the media would acidify? From the context of the paragraph I am assuming that your are talking about cancer cells in vivo; therefore a different term to 'media' would be more appropriate.

We thank the reviewer for this comment. We clarified the description of this study, which shows the effects of colorectal cancer cells adapted to extracellular acidosis in vitro (media), as well as when these acidosis-adapted cells are grown in vivo in tumor xenographs (Corbett et al., Cell Metabolism 2016, reference 22). The paragraph reads as follows in line 216:

“Accordingly, FAO can even support anabolism in Warburg-abiding colorectal cancer cells, but only as an adaptative response to extracellular acidification [22]. This increase in FAO was even preserved when these acidosis-adapted cancer cells generated in vitro were grown as tumor xenographs in mice [22].”

2) Please define the abbreviations in Figure 1 and Figure 2, notably FASN and PDM.

Thank you. We have defined all abbreviations in the figure legends, as well as in the text. FASN is the gene annotation for fatty acid synthase and PDM for peridroplet mitochondria.

3) Line 84: do you mean escape?

Thank for spotting this typographical error. Yes, we meant escape

4) Please check spelling/grammar at Lines 96, 112, 114, 121/122, 124, 129, 151, 255, 274, 279, 307, 313, 315.

We have done an extensive revision of the entire manuscript to correct grammar and spelling, including the lines stated by the reviewer.

5) Line 234: do you mean 'form of' instead of 'from'?

Thank for identifying this typographical error. Indeed, we meant “form of”. Corrected in now line 336.

Reviewer 2 Report

In this manuscript, Pinto De Oliveira and Liesa review the role of lipid metabolism, mitochondrial beta oxidation in particular, in the proliferation and metastatic potential of tumour cells. This topic is interesting and important, and I was looking forward to learn more about it. Because of general lack of precision and clarity on metabolic biochemistry, however, the manuscript turned out rather hard to access, and I was left confused, even after repeated reading, as to the main points the authors are trying to make. The following examples are not exhaustive, but illustrate my confusion.

1.    The definition of glycolysis is imprecise: both anaerobic and aerobic fates of pyruvate require glucose breakdown. This imprecision extends to the term ‘pyruvate utilisation’. For instance, is it meant that pyruvate reduction to lactate is lowered in an acidic environment, or that pyruvate oxidation/carboxylation are also compromised?

2.    The statement on the tumour cells’ ‘addiction to glucose’ for generation of nucleic acids and  amino acids needs clarification. It may be clear to a general readership that the pentose phosphate pathway is driven by glucose-6-phosphate. The glucose need of amino acid synthesis is less obvious, at least to me. Alpha-ketoglutarate is transformed to glutamate via reductive amination (catalysed by glutamate dehydrogenase), which then participates in transamination reactions to turn pyruvate, oxaloacetate (etc) in amino acids. In other words, TCA cycle intermediates and TCA cycle precursors are the building blocks for most non-essential amino acids. It took me a while to remember that accumulation of TCA metabolites requires carboxylation of glucose-derived pyruvate to supply carbon to the TCA cycle by anaplerosis. Confusing matters down the line, it is also argued that such carbon is obtained from glutamine in cancer cells, making it less clear why tumour cells should be ‘addicted to glucose’.

3.    Suggesting that ‘active’ gene upregulation by oncogenes and loss of tumour suppressors implies that increased glucose use (by anaerobic glycolysis) cannot be a ‘passive’ response to elevated proliferation is confusing – what is meant by ‘passive’ and ‘active’ in this context? There will always be some need for a signal to trigger gene expression.

4.    Nutrients are not only needed to make building blocks for proliferation-related biosynthesis, i.e., to support anabolism (as is suggested implicitly), but also to fuel biosynthesis by maintaining a high phosphorylation potential through catabolism. This interconnection between different parts of metabolism should be explained more clearly.

5.    What exactly is meant by ‘redox stress’? Given the high energetic needs of a proliferating cell, how likely is it that tumour cells become so reduced that generation of reactive oxygen species becomes excessive?

6.    The authors speculate that simultaneous breakdown and synthesis of fatty acids, as seen in certain cancers, might be possible due to the presence of distinct mitochondrial populations. They have shown the existence of such subcellular compartmentalisation in brown adipose tissue where one pool of mitochondria associates with lipid droplets and is only involved in glucose oxidation, whereas another cytosolic pool oxidises fatty acids. Such mitochondrial segregation is entirely speculative for cancer cells, and would pose various questions. For instance, where do the ‘cytosolic’ mitochondria source there fatty acids if they are needed to expand lipid droplets? How is pyruvate access to these mitochondria prevented? How is diffusion of citrate/malonyl-CoA between different mitochondrial pools prevented?

7.    The apparent need for an energetically futile cycle of fatty acid oxidation and (glutamine-fuelled) fatty acid synthesis remains unexplained. Such cycling is counter-intuitive given the exceptionally high energy demand of fast-growing lymphomas, the subset of cancers in which fatty acids are burnt against a high-glucose background. The antioxidant defence argument (lines 76-78) is equally unsatisfactory. How would mitochondrial beta oxidation be required for glutathione reduction? Are the generated reducing equivalents not the very cause of increased reactive oxygen species production?

8.    The idea that certain tumours (lymphomas) ‘escape’ or ‘challenge’ the Randle cycle by burning both fat and sugar simultaneously is not as surprising as the authors seem to think. It is fairly well established that Randle-like regulation of fuel selection is overruled under conditions of high energy demand (http://dx.doi.org/10.1042/bst0311157). As proliferating tumour cells have a high energy demand, it is perhaps more surprising that most cancer cells do indeed obey the Randle cycle, likely as a consequence of the Warburg shift towards anaerobic metabolism. In this respect, is the question not why lymphomas do not exhibit a Warburg effect?

9.    Organisation of content is confusing, as the manuscript seemingly at will switches from ‘normal’ cancer descriptions to discussion of the exceptional (Randle-defying) subsets. Use of language needs attention throughout.

Author Response

Reviewer # 2:

1) The definition of glycolysis is imprecise: both anaerobic and aerobic fates of pyruvate require glucose breakdown. This imprecision extends to the term ‘pyruvate utilisation’. For instance, is it meant that pyruvate reduction to lactate is lowered in an acidic environment, or that pyruvate oxidation/carboxylation are also compromised?

We substituted “pyruvate utilization” for “pyruvate oxidation in mitochondria” or “pyruvate reduction to lactate” to increase precision. The particular study discussed (reference 22) demonstrated that both pyruvate oxidation and reduction were decreased by acidosis, as well as pyruvate synthesis by glycolysis. To clarify this point, we have extensively rewritten the paragraph describing this study (line 216): “Accordingly, FAO can even support anabolism in Warburg-abiding colorectal cancer cells, but only as an adaptative response to extracellular acidification [22]. This increase in FAO was even preserved when these acidosis-adapted cancer cells generated in vitro were grown as tumor xenographs in mice [22]. High glycolytic rates can acidify the cytosol and the extracellular space via the generation of lactate, meaning that accumulation of acid can cause product-mediated inhibition of glycolysis. As a result, this shift to FAO is thought to be a response to an acidosis-induced decrease in glycolysis to rates that cannot sustain proliferation. In this regard, Corbett et al., observed that acidosis caused a decrease in glucose consumption and pyruvate oxidation in the mitochondria, pointing to a reduction in pyruvate synthesis by glycolysis as well [22].”

2) The statement on the tumour cells’ ‘addiction to glucose’ for generation of nucleic acids and amino acids needs clarification. It may be clear to a general readership that the pentose phosphate pathway is driven by glucose-6-phosphate. The glucose need of amino acid synthesis is less obvious, at least to me. Alpha-ketoglutarate is transformed to glutamate via reductive amination (catalysed by glutamate dehydrogenase), which then participates in transamination reactions to turn pyruvate, oxaloacetate (etc) in amino acids. In other words, TCA cycle intermediates and TCA cycle precursors are the building blocks for most non-essential amino acids. It took me a while to remember that accumulation of TCA metabolites requires carboxylation of glucose-derived pyruvate to supply carbon to the TCA cycle by anaplerosis. Confusing matters down the line, it is also argued that such carbon is obtained from glutamine in cancer cells, making it less clear why tumour cells should be ‘addicted to glucose’.

We thank the reviewer for giving us the opportunity to clarify such an important concept. We expanded our discussion with the specific evidence supporting that tumors selected glycolysis to lactate not just because of faster ATP provision, but because of fast provision of carbon intermediates for anabolism. As an important example, we discussed one study demonstrating how glycolytic intermediates, which are precursors of pyruvate, fueled aminoacid and nucleotide synthesis (line 186):

“Recent studies demonstrate that glycolytic intermediates and precursors of pyruvate, such as 3-phosphoglycerialdehide, are needed as a carbon source to fuel the synthesis of  amino and nucleic acids at the high rates imposed by tumor proliferation [11,12]. Moreover, glucose also fuels the pentose phosphate pathway [13], which is needed to synthesize nucleotides, generate NADPH for lipid biosynthesis and to counteract elevated ROS production caused by the loss of tumor suppressors or by stressors of the tumor environment  [14,15]. In addition, not all pyruvate generated by glycolysis is used to produce lactate. A fraction of glucose-derived pyruvate is diverted to mitochondria to produce citrate and fuel FA and lipid biosynthesis, with about 60% of fatty acyl carbons are derived from glucose in glioblastoma [16].”

We expanded the discussion on how PKM2 expression in cancer support that the contribution of glycolysis to anabolism goes beyond pyruvate being carboxylated in the mitochondria for anaplerosis (lines 288-295).

3) Suggesting that ‘active’ gene upregulation by oncogenes and loss of tumour suppressors implies that increased glucose use (by anaerobic glycolysis) cannot be a ‘passive’ response to elevated proliferation is confusing – what is meant by ‘passive’ and ‘active’ in this context? There will always be some need for a signal to trigger gene expression.

We apologize for the lack of precision in the term “passive”. We have clarified the message we wanted to deliver with the following text in line 179.

“Initially, it was proposed that the Warburg effect was an indirect consequence of proliferation, namely an adaptive response to the exceedingly high ATP demand imposed by the augmented rates of cell division. However, the relatively recent demonstration that the transcription of genes encoding glycolytic enzymes is directly upregulated by the action of oncogenes, by the loss of genes encoding tumor suppressors and by kinases that also signal proliferation, support that higher glycolytic rates in tumors are not simply a secondary or adaptive response to elevated ATP demand (or defined by others in the literature as a “passive response”).”

4) Nutrients are not only needed to make building blocks for proliferation-related biosynthesis, i.e., to support anabolism (as is suggested implicitly), but also to fuel biosynthesis by maintaining a high phosphorylation potential through catabolism. This interconnection between different parts of metabolism should be explained more clearly.

We are thankful to the reviewer for this suggestion. We have clarified this interconnection, highlighting the concept that high phosphorylation potential is only prioritized by tumors under stressful and catabolic conditions. On the other hand, relatively unstressed tumors select glycolysis because it is more efficient sustaining anabolism. The reason is that FAO has a higher capacity to maintain a higher phosphorylation potential per molecule of nutrient than glucose, and can provide ATP as fast as required for cancer cell proliferation, as reference 22 shows. This concept is included in the review in different sections:

       Line 212: “The preference for glucose over fatty acids to sustain cancer cell proliferation supports that the need for efficient ATP provision is not the main driver for glucose preference in cancer. FAO is more efficient generating ATP per molecule of nutrient than glucose-derived pyruvate oxidation. Moreover, more recent studies demonstrate that FAO can achieve rates to generate ATP as fast as required to support cancer proliferation [22].”

       Line 238: “If FAO can support high proliferation rates in cancer, why colorectal cancers choose glycolysis instead of FAO combined with glutamine oxidation to support proliferation under non-stressed conditions? The reason might be that FAO does not provide a selective advantage and introduces a metabolic liability that decreases cancer cell aggressiveness.”

Line 293: “These properties of PKM2 enzymatic activity highlight that the major driver for glycolysis remodeling in cancer might be increasing anabolic capacity, rather than ATP synthesis.“

5) What exactly is meant by ‘redox stress’? Given the high energetic needs of a proliferating cell, how likely is it that tumour cells become so reduced that generation of reactive oxygen species becomes excessive?

Thank you for this comment. We have substituted the “redox stress” term with increased ROS production. In our review, we are referring to the three major cancer-enriched processes that increase ROS production: 1) Loss of tumor suppressor genes, as some of the proteins encoded by these genes block ROS. 2) Cellular detachment from the matrix needed for metastasis causes an increases in ROS production. 3) FA oxidation and glucose restriction can increase ROS production, by different mechanisms. These have been appropriately cited in lines 256, 306, 326, 343, 500.

6) The authors speculate that simultaneous breakdown and synthesis of fatty acids, as seen in certain cancers, might be possible due to the presence of distinct mitochondrial populations. They have shown the existence of such subcellular compartmentalisation in brown adipose tissue where one pool of mitochondria associates with lipid droplets and is only involved in glucose oxidation, whereas another cytosolic pool oxidises fatty acids. Such mitochondrial segregation is entirely speculative for cancer cells ,and would pose various questions. For instance, where do the ‘cytosolic’ mitochondria source there fatty acids if they are needed to expand lipid droplets? How is pyruvate access to these mitochondria prevented? How is diffusion of citrate/malonyl-CoA between different mitochondrial pools prevented?

We thank the reviewer for raising these questions, which greatly helped to refine our hypothetical model. As a result of addressing reviewer 3 concerns of the need to add the enzymes determining the Randle cycle, we answered these questions by:

  • We renamed peridroplet mitochondria as lipogenic mitochondria for cancer cells. The reason is that lipid synthesis driven by mitochondria, as we found in BAT, can be independent of the existence of large lipid droplets. Indeed, many lipogenic tumors that depend on mitochondria to synthesize FA do not have lipid droplets.

  • We speculate that ACC2 might not be present in every single mitochondrion, with ACC2 being only present or enriched selectively in lipogenic mitochondria. We also speculate that the two isoforms of CPT1, a and b, which have different sensitivities to malonyl-CoA, might be differentially distributed among these two populations of mitochondria (see new Figure 2). Briefly, we propose that mitochondria oxidizing FAO will carry CPT1-a and not ACC2, making FAO-mitochondria more insensitive to FAO-inhibition mediated by malonyl-CoA (lines 453-458). On the other hand, lipogenic mitochondria will carry CPT1-b and ACC2, making them highly sensitive to malonyl-CoA-mediated inhibition of FAO and hubs for malonyl-CoA synthesis (lines 447-453). In addition, lipogenic mitochondria, as peridroplet mitochondria in BAT, can oxidize more pyruvate and produce more citrate to generate more malonyl-CoA.

  • This lipid droplet-independent model answers the question that the source of FA can be the extracellular space, as shown by the two references showing concurrency of FAO and FA synthesis (Caro et al., 2012, reference 25; Corbett et al., 2016, reference 22). The importance of extracellular nutrients in this model is discussed not in lines 419-444.

7) The apparent need for an energetically futile cycle of fatty acid oxidation and (glutamine-fuelled) fatty acid synthesis remains unexplained. Such cycling is counter-intuitive given the exceptionally high energy demand of fast-growing lymphomas, the subset of cancers in which fatty acids are burnt against a high-glucose background. The antioxidant defence argument (lines 76-78) is equally unsatisfactory. How would mitochondrial beta oxidation be required for glutathione reduction? Are the generated reducing equivalents not the very cause of increased reactive oxygen species production?

We agree with the reviewer that ATP and antioxidant activity might not be the most important process driving the selective advantage of using FAO, for the same reason that the reviewer pointed: oxidizing a FA that was just synthesized can be a cycle wasting ATP and NADPH. The conclusion that FAO was recruited to cover the exceptionally high demand for ATP and NADPH was drawn by the article characterizing these lymphomas, with experimental evidence supporting it (Caro et al., Cancer Cell 2012, reference 25).

We propose in our review that FAO might provide an anabolic advantage in these tumors that justify wasting some ATP and NADPH in this FA-FAO futile cycle, similarly to what was described for the Warburg effect (i.e. ATP synthesis is not the main driver for glycolysis, but anabolism and the synthesis of building blocks). Furthermore, we believe that the anabolic advantage associated with FAO can be explained better with the existence of two segregated mitochondria populations (see response to point 6). Briefly, the co-existence of a segregated population of anabolic/lipogenic mitochondria would allow using pyruvate-derived from glycolysis selectively for anaplerosis and fuel anabolism, which would preserve glutamine for other purposes. The other population oxidizing FAO can provide more ATP and NADPH per molecule of nutrient, and probably other uncharacterized intermediate(s) that favor proliferation in these tumors.

 8) The idea that certain tumours (lymphomas) ‘escape’ or ‘challenge’ the Randle cycle by burning both fat and sugar simultaneously is not as surprising as the authors seem to think. It is fairly well established that Randle-like regulation of fuel selection is overruled under conditions of high energy demand (http://dx.doi.org/10.1042/bst0311157). As proliferating tumour cells have a high energy demand, it is perhaps more surprising that most cancer cells do indeed obey the Randle cycle, likely as a consequence of the Warburg shift towards anaerobic metabolism. In this respect, is the question not why lymphomas do not exhibit a Warburg effect?

We appreciate the reviewer raising this point, which illustrated that we did not communicate the gap in knowledge appropriately. Indeed, the reference cited by the reviewer demonstrates that when the ATP levels drop and AMPK is activated, both glycolysis and FAO can concurrently occur to bring ATP to normal levels. Indeed, this is the main argument used by reference 25 to justify the concurrency of glycolysis, glycolysis-driven FA synthesis and FAO. However, AMPK activation leads to the inhibition of both ACC1 and ACC2, which blocks malonyl-CoA synthesis. Blockage of malonyl-CoA synthesis impedes the use of citrate to synthesize FA, which could cause citrate accumulation and thus glycolysis blockage. Thus, the surprise is coming from:

  • The increase ATP demand, with the concomitant drop in ATP levels and AMPK activation cannot explain the phenotype of these tumors, as they show increased FA synthesis. Thus, the escape from the Randle cycle cannot be explained by AMPK activation.

  • This result per se strongly suggests that an elevation in ATP demand is not the main driver for increased FAO in these lymphomas, otherwise FA synthesis could not occur.

In regards for these tumors not showing Warburg, these tumors still show increased lactate production. The main finding by Caro et al., 2012 (reference 25) is that FAO occurs on top of glycolysis to lactate, rather that Warburg is not occurring per se in these tumors.

9) Organisation of content is confusing, as the manuscript seemingly at will switches from ‘normal’ cancer descriptions to discussion of the exceptional (Randle-defying) subsets. Use of language needs attention throughout.

We extensively revised the organization of the manuscript. We now include a new first section that defines the Randle cycle in healthy tissues and lists the main genes and processes involved in the cycle (lines 35-121, including one Figure). We added a paragraph discussing the gaps in knowledge in the regulation of glycolysis and FAO competition in health (lines 123-173). This new first section is now followed with the most common features of metabolic remodeling in cancer, describing how colorectal cancer and glioblastoma show the Warburg effect and abide the Randle cycle (lines 174-208). Then, we provide an introductory paragraph summarizing how Randle abiding tumors only recruit FAO under stress and compare them with the tumors that use FAO to proliferate, raising important questions on how non-Randle tumors can execute both FAO and glycolysis (new Table 1, lines 210-283). We then move to a subchapter describing how glycolytic remodeling in tumors is compatible with FAO blocking glycolysis, while metastasis can upregulate FAO as a response to stress or glucose deprivation (Lines 284-382). We then discuss the mechanisms described to date that explain the existence of non-Randle abiding tumors (lines 383-406). After this section, we propose our hypothesis on a new mechanism by which tumors can escape the Randle cycle (lines 407-465, new Figure 2).  Finally, we discuss how targeting FAO in cancer and changing extracellular nutrient availability, recapitulating the Randle Cycle, could be exploited to stop cancer growth (lines 466-580, including new Figure 3).

Reviewer 3 Report

Ms. Ref. No. cells-971037

Title: The role of mitochondrial fat oxidation in cancer cell proliferation

The authors reviewed that some cancer types have a Randle cycle-abiding behavior, while B-cell lymphoma and leukemias escape the Randle cycle. The authors also discussed several potential FAO-related therapeutic strategies. Overall, the authors may want to improve the manuscript by providing a clear logical flow with figures that provide more detailed information for readers to better understand the topic.

Major concerns:

  1. For the first part, the authors reviewed glucose, glutamine, and FAO in glioblastoma and colorectal cancer. The authors should include a table to summarize the metabolic similarities and differences across these cancer types and explicitly summarize the major knowledge gained from these studies.
  2. For the second part, the authors should first introduce the Randle cycle under normal conditions in detail with figures pointing out the key enzymes mentioned in the manuscript, and then review the cancer types that follow the Randel cycle. Lastly, the authors should review the cancer types that do not follow the Randle cycle. Thus, Figure 1 and part 2 need to be extensively revised.
  3. The authors may also want to expand Figure 2 by adding key enzymes mentioned in the manuscript so the readers will have a better understanding of the manuscript.
  4. The authors briefly touched on some FAO-related parts without clear explanations. For example, what the role of tumor suppressor NKX2-8 in ovarian cancer is, and why ablation of NKX2-8 activates FAO, and through what kind of mechanism are all points that need to be elaborated on. The authors only quickly scratched the surface, which does not benefit the readers.
  5. For the Integrated Stress Response in part 4, the authors may want to introduce the genes involved in the pathway and provide a figure.

Overall, the authors should focus on the key genes or pathways involved in FAO and explicitly explain their roles in different cancer types.

Author Response

Reviewer # 3:

1) For the first part, the authors reviewed glucose, glutamine, and FAO in glioblastoma and colorectal cancer. The authors should include a table to summarize the metabolic similarities and differences across these cancer types and explicitly summarize the major knowledge gained from these studies.

We thank the reviewer for this comment. We transformed previous Figure 2 into a new Table 1 (line 259). This table summarizes the metabolic similarities of glioblastoma and colorectal cancers, as cancers that abide to the Warburg effects and the Randle-cycle. We added a row with a second category of non-Randle cycle abiding tumors and how their metabolism differs from the first category.  

2) For the second part, the authors should first introduce the Randle cycle under normal conditions in detail with figures pointing out the key enzymes mentioned in the manuscript, and then review the cancer types that follow the Randel cycle. Lastly, the authors should review the cancer types that do not follow the Randle cycle. Thus, Figure 1 and part 2 need to be extensively revised.

We thank the reviewer for such an important comment. We wrote a new section, which became new part 1, that defines in detail the Randle cycle (lines 35-173, including one Figure). This new section now lists the enzymes and the key regulatory processes involved in the Randle cycle. In addition, new Table 1 summarizes which tumors abide to the Randle cycle and the studies demonstrating this (line 259).

3) The authors may also want to expand Figure 2 by adding key enzymes mentioned in the manuscript so the readers will have a better understanding of the manuscript.

We have addressed this important point. New Figures 1 (line 58) and 2 (line 423) now include the key enzymes involved in the Randle cycle, as well as in fatty acid oxidation and synthesis.

4) The authors briefly touched on some FAO-related parts without clear explanations. For example, what the role of tumor suppressor NKX2-8 in ovarian cancer is, and why ablation of NKX2-8 activates FAO, and through what kind of mechanism are all points that need to be elaborated on. The authors only quickly scratched the surface, which does not benefit the readers.

We thank the reviewer for identifying this issue. With this study, we wanted to highlight that FAO can be determined by transcriptional programs related to tumor suppressors as well, which can be completely unrelated to changes in nutrient availability or decreased anti-ROS function. Thus, NKX2-8 expression can be used as a biomarker for tumors that might be responsive to FAO inhibition. It is important expanding this section and the text reads now as (lines 490):

“Ovarian cancer patients with low expression of the tumor suppressor NKX2–8 have a poor prognosis, with the deletion of NKX2-8 activating FAO and increasing chemoresistance [64]. NKX2-8 (Homeobox protein Nkx-2.8) is a transcription factor regulating development. The loss of NKX2-8 dissembles a deacetylation complex, resulting in chromatin acetylation leading to increased CPT1-A and CPT2 expression in ovarian cancer [64]. Adipocytes were shown to provide a favorable environment for the growth of these NKX2-8 deleted ovarian tumors, as adipocytes retained chemotherapeutic reagents and provided fatty acids to fuel FAO [64]. Thus, blocking FAO or even decreasing fatty acid availability could be a strategy to stop ovarian tumor growth.”

5) For the Integrated Stress Response in part 4, the authors may want to introduce the genes involved in the pathway and provide a figure.

We thank the reviewer for raising this point. We now provide a new Figure 3 summarizing the ISR (line 559), as well as expanding the discussion on the role of key ISR regulators OMA1, DELE1, HRI and ATF4 (lines 531- 550).

Overall, the authors should focus on the key genes or pathways involved in FAO and explicitly explain their roles in different cancer types.

We have now focused on the key genes involved in FAO that explained the Randle cycle (CPT1 and ACC1/2) and defined their importance in colorectal cancer and lymphomas, as they are the tumors showing the most divergent metabolic behaviour in terms of FAO. We want to thank this reviewer for the time s/he dedicated to revise our manuscript, making it substantially better.

Round 2

Reviewer 3 Report

The authors addressed my previously raised concerns and improved the manuscript. Minor language changes may be required.